# Low Prognostic Nutritional Index Predicts In-Hospital Complications and Case Fatality in Patients with Spontaneous Intracerebral Hemorrhage: A Retrospective Study

**DOI:** 10.3390/nu16121841

**Published:** 2024-06-12

**Authors:** Shang-Wun Jhang, Yen-Tze Liu, Chew-Teng Kor, Yi-Ping Wu, Cheng-Hung Lai

**Affiliations:** 1Department of Veterinary Medicine, College of Veterinary Medicine, National Chung Hsing University, Taichung 402, Taiwan; 133393@cch.org.tw; 2Department of Neurosurgery, Changhua Christian Hospital, Changhua 500, Taiwan; 183464@cch.org.tw; 3Department of Family Medicine, Changhua Christian Hospital, Changhua 500, Taiwan; 144084@cch.org.tw; 4Big Data Center, Changhua Christian Hospital, Changhua 500, Taiwan; 179297@cch.org.tw; 5Department of Post-Baccalaureate Medicine, College of Medicine, National Chung Hsing University, Taichung 402, Taiwan; 6Graduate Institute of Statistics and Information Science, National Changhua University of Education, Changhua 500, Taiwan

**Keywords:** prognostic nutrition index (PNI), spontaneous intracerebral hemorrhage (ICH), in-hospital complication, case fatality, propensity score matching (PSM)

## Abstract

Background: Spontaneous intracerebral hemorrhage (ICH) is associated with high case fatality and significant healthcare costs. Recent studies emphasize the critical role of nutritional status in affecting outcomes in neurological disorders. This study investigates the relationship between the Prognostic Nutrition Index (PNI) and in-hospital complications and case fatality among patients with ICH. Methods: A retrospective analysis was performed using data from the Changhua Christian Hospital Clinical Research Database between January 2015 and December 2022. Patients under 20 or over 100 years of age or with incomplete medical data were excluded. We utilized restricted cubic spline models, Kaplan–Meier survival analysis, and ROC analysis to assess the association between PNI and clinical outcomes. Propensity score matching analysis was performed to balance these clinical variables between groups. Results: In this study, 2402 patients with spontaneous ICH were assessed using the median PNI value of 42.77. The cohort was evenly divided between low and high PNI groups, predominantly male (59.1%), with an average age of 64 years. Patients with lower PNI scores at admission had higher in-hospital complications and increased 28- and 90-day case fatality rates. Conclusions: Our study suggests that PNI could serve as a valuable marker for predicting medical complications and case fatality in patients with spontaneous ICH.

## 1. Introduction

Intracerebral hemorrhage (ICH) represents over 10% of the approximately 17 million strokes that occur globally each year. The case fatality rate for ICH exceeds 40%, and merely 20% of those cases who survive achieve functional independence within six months [1,2]. In Taiwan, cerebrovascular diseases have been ranked fifth among the top ten causes of death in 2022. According to Taiwanese stroke registry data, intracerebral hemorrhage (ICH) makes up 15.9% of all stroke cases. It is linked to a 30-day case fatality rate of 19.8% and a 1-year case fatality rate of 29.6% [3]. The incidence of ICH is higher in Taiwan compared to Western populations [1,2,3]. An analysis of the Taiwan National Health Insurance Research Database from 2011 to 2014 revealed that the odds of death for those with ICH were significantly higher, with an adjusted odds ratio of 5.80 compared to 2.07 for ischemic stroke [4]. Spontaneous intracerebral hemorrhage (ICH) remains one of the most serious subtypes of stroke, characterized by a high case fatality rate, higher medical costs, and often resulting in poor functional outcomes [3,4].

Nutritional status is associated with case fatality, length of hospitalization, and clinical outcomes with cerebrovascular diseases [5,6,7]. Previous research has indicated that acute ICH patients with poor outcomes tend to exhibit a greater degree of undernutrition compared to those with better outcomes [8]. A low level of serum albumin can increase the risk of infections and case fatality in critically ill patients [9,10]. Hypoalbuminemia is a simple marker of malnutrition that is a major cause of impaired immune response and serves as a reliable predictor of hospital-acquired infections [11]. Previous reports suggested a relationship between hypoalbuminemia and increased case fatality in patients with ICH [9,12]. Furthermore, admission lymphopenia is associated with an increased risk of infection and unfavorable outcomes in patients with spontaneous ICH [10].

The Prognostic Nutritional Index (PNI) serves as a practical and objective prognostic factor. It assesses nutritional and immunological statuses by measuring serum albumin levels and the count of peripheral blood lymphocytes. Consequently, PNI is a viable tool for clinical application. Recent research has indicated that lower PNI values correlate with poorer outcomes in patients classified under the New York Heart Association’s guidelines for coronary heart disease [11]. Yang et al. found that PNI was an independent predictor of 30-day, 90-day, and 1-year case fatality in critically ill patients that experienced stroke [13]. An observational study revealed that preoperative PNI could be associated with the occurrence of postoperative pneumonia in patients with aneurysmal subarachnoid hemorrhage (aSAH) [14].

The link between the Prognostic Nutritional Index (PNI) and outcomes for patients with spontaneous intracerebral hemorrhage (ICH) has yet to be documented. Our research thus focuses on investigating this relationship. Based on prior evidence, we hypothesized that a lower PNI correlates strongly with increased all-cause morbidity and case fatality among patients with spontaneous ICH.

## 2. Materials and Methods

### 2.1. Study Populations

This retrospective study was conducted between January 2015 and December 2022, including all patients admitted for spontaneous intracerebral hemorrhage (ICH) who received medical or surgical treatment in our medical unit at Changhua Christian Hospital, Taiwan. The diagnosis of ICH was initially confirmed by brain computed tomography (CT) scans at admission. The institutional review board waived the requirement of informed consent, since no intervention was performed, and no personally identifiable information was disclosed.

### 2.2. Inclusion/Exclusion Criteria

Patients were included if they had spontaneous intracerebral hemorrhage. We excluded (1) patients <20 years, (2) patients >100 years, (3) patients with missing vital sign data, or (4) patients with incomplete biochemical data from this study. We compared the demographics of patients who were excluded due to incomplete vital sign and biochemical data (Appendix A).

### 2.3. Data Collection and Definition

We included 5633 patients diagnosed with spontaneous intracerebral hemorrhage (ICH) at Changhua Christian Hospital from January 2015 to December 2022. These patient data were sourced from the Changhua Christian Hospital Clinical Research Database (CCHRD). Ultimately, our study comprised 2402 ICH cases (as illustrated in Figure 1). Approval for this study was granted by the Institutional Review Board of Changhua Christian Hospital (IRB no: 230539).

We collected the following data from medical records: demographic variables (age, gender, and body mass index (BMI), calculated as weight in kilograms divided by height in meters squared), comorbidities (hypertension, diabetes mellitus, and dyslipidemia), and vital signs at admission (temperature, heart rate, respiratory rate, systolic and diastolic blood pressures, and percutaneous oxygen saturation (SpO2). Laboratory parameters included counts of red and white blood cells, lymphocytes, hemoglobin concentration, platelet count, red blood cell distribution width (RDW), and levels of serum creatinine, blood urea nitrogen (BUN), serum potassium, serum sodium, albumin, and HbA1c. The Charlson Comorbidity Index, which estimates survival based on 17 comorbidity-related items, was used to assess the impact of comorbid conditions on case fatality. The PNI was calculated as follows: PNI = albumin (g/L) + 5 × lymphocyte (10^9^/L). Patients were divided into low-value groups (PNI < 42.77) and high-value groups (PNI ≥ 42.77) according to the cut-off point.

### 2.4. Study Results and Definitions

The primary outcome was medical complications that developed during hospitalization, such as acute respiratory failure, stroke-associated pneumonia (SAP), urinary tract infection (UTI), seizures, cardiopulmonary resuscitation (CPR), sepsis, gastrointestinal bleeding, and acute venous thromboembolism, as recorded in the forms of the paper registry. Secondary outcomes were case fatality at 28 days and 90 days after ICH. The start date was the date of admission of the patient.

### 2.5. Statistical Analysis

Categorical variables were presented as numerical proportions and continuous variables as mean ± SD or median (IQR) depending on their distribution. Student’s *t*-test or Mann–Whitney U test was used for continuous variables and chi-square test for categorical variables. Confounding factors always diminish the accuracy and objective of retrospective studies. Propensity score matching (PSM) analysis was performed to balance these clinical variables between groups, which could reduce potential bias to some extent [15]. This involved matching patients 1:1 using the nearest-neighbor algorithm, with a caliper width of 0.017 and without replacement, to discern PNI’s effect on ICH outcomes. Furthermore, PSM adjusted for clinical parameters with significant *p*-values (<0.05) in the multivariate analysis.

To explore the relationship between medical complications and PNI levels, logistic regression models were utilized. The association between PNI levels and 28- and 90-day case fatality rates was analyzed using crude and adjusted Cox’s proportional hazard models. Two adjustment models were applied before and after propensity score matching to strengthen result reliability.

Multivariate regression analyses were adjusted for potential covariates. The impact of PNI levels on the odds of in-hospital complications and the hazard of case fatality at 28 and 90 days was analyzed using a restricted cubic spline with four knots. Survival outcomes were examined through Kaplan–Meier curves and assessed using log-rank tests, which identify differences in survival experiences among groups. The predictive accuracy of PNI was evaluated by the ROC curve, presenting results as the AUC. A *p*-value of <0.05 was considered statistically significant in this study.

All statistical analyses were performed with SAS 9.4 version and R software (version 4.1.0) and the R Archive Network (http://cran.rproject.org, accessed on 18 May 2021). In this study, we define a *p*-value below 0.05 as statistically significant.

## 3. Results

### 3.1. Patient Characteristics before and after Propensity Score Matching

Among the 5633 patients initially enrolled with spontaneous intracerebral hemorrhage (ICH), we excluded 3231 patients due to age being outside the 20 to 100 years range (1091 patients), missing vital sign data (1354 patients), or missing biochemical data (786 patients). Ultimately, 2402 patients meeting the screening criteria were included in the study (Figure 1). The reference median PNI value was set at 42.77 (OR = 1). The comparison of demographic and clinical characteristics between the low PNI (<42.77) and high PNI (≥42.77) groups before and after the propensity score matching is presented in Table 1.

Among the 2402 patients included in the present study, 1201 (50%) were in the low PNI group and 1201 (50%) in the high PNI group, and the study population was 59.1% male, and the mean age at diagnosis was 64 years. Before propensity score matching, patients with spontaneous ICH and PNI < 42.77 had lower levels of BMI (*p* = 0.004), GCS scales (*p* < 0.001), diastolic blood pressure (*p* = 0.039), body temperature *p* = 0.023), Hb (*p* < 0.001), Na (*p* < 0.001), RBC (*p* < 0.001), albumin (*p* < 0.001), platelet (*p* < 0.001), and lymphocyte counts (*p* < 0.001) and higher levels of age (*p* < 0.001), systolic blood pressure (*p* < 0.001), heart rate (*p* < 0.001), respiratory rate (*p* = 0.006), creatine (*p* < 0.001), K (*p* = 0.02), blood urea nitrogen (*p* < 0.001), red blood cell distribution width (*p* < 0.001), Charlson Comorbidity Index (*p* < 0.001), antihypertension treatment (*p* < 0.001), non-insulin antihyperglycemic treatment (*p* = 0.036), and insulin treatment (*p* = 0.001). Subsequent propensity score matching resulted in two groups: low PNI (*n* = 845) and high PNI (*n* = 845), achieving balance across all covariates. Statistically significant differences remained between groups in BMI, GCS, Hb, RDW, albumin, white blood cells (WBCs), and lymphocyte counts (Table 1).

### 3.2. Association between PNI and In-Hospital Complication

In this study, patients were divided into two groups based on their PNI values: <42.77 (*n* = 1201) and ≥42.77 (*n* = 1201). Table 2 presents the association between PNI and in-hospital complications. Univariate logistic regression analysis indicated that the high PNI group had a significantly reduced risk of in-hospital complications, with an odds ratio (OR) of 0.64 (95% CI: 0.53–0.77; *p* < 0.001), suggesting a 36% reduction in risk compared to the low PNI group. This finding was consistent in the multivariate logistic regression model, where the high PNI group had an OR of 0.69 (95% CI: 0.55–0.87; *p* = 0.002). Furthermore, after adjusting for potential confounders through propensity score matching, the OR for in-hospital complications in the high PNI group was 0.74 (95% CI: 0.58–0.94; *p*-value = 0.013). These results underscore a significant association between higher PNI values and a lower risk of in-hospital complications, as demonstrated in both pre- and post-propensity score-matched analyses.

Restricted cubic spline analyses illustrate the shapes of the multivariate association between the PNI and in-hospital complications in the single-center study. An elevated risk of complications was observed with decreasing PNI levels (Figure 2).

### 3.3. Association between PNI and 28- and 90-Day Case Fatality

We performed a Cox regression analysis to explore the association between PNI and case fatality in patients with spontaneous ICH. The results, presented in Table 3, revealed that the PNI serves as an independent predictor of 28- and 90-day case fatality in patients with spontaneous ICH. In the unadjusted model, the hazard ratio for the high group (PNI ≥ 42.77) was 0.46 (95% CI: 0.36–0.59) and 0.45 (95% CI: 0.35–0.57), respectively, that is, the decreased PNI (PNI < 42.77) was significantly associated with an increased incidence of 30-day and 90-day case fatality. After adjustment was made, the hazard ratio (of 30-day and 90-day case fatality for the high group) was 0.72 (95% CI: 0.55–0.94) and 0.72 (95% CI: 0.56–0.93), respectively, compared to the lower group. In the multivariate logistic regression analysis after PSM, the HR (of 30-day and 90-day case fatality) of the PNI  ≥ 42.77 group was 0.71 (95% CI: 0.52–0.98) and 0.69 (95% CI: 0.51–0.94), respectively (Table 3).

We further examined the crude hazard ratio (HR) for 90-day case fatality by treating the PNI as a continuous variable, utilizing restricted cubic spline analysis in the Cox model, as shown in Figure 3. The figure illustrates that the increased PNI level in the restricted cubic spline curve showed a trend of monotonic decrease in the risk of 90-day all-cause case fatality.

### 3.4. Risk Factor for in-Hospital Complication and 28- and 90-Day Case Fatality

In our analysis of risk factors, higher pulse rate, increased body temperature, and nasogastric (NG) tube insertion were associated with an increased risk of in-hospital complications (Figure 4A). Conversely, higher Glasgow Coma Scale (GCS) scores, red blood cell (RBC) count, diastolic blood pressure (DBP), and heart rate were linked to a reduced risk of in-hospital complications. Regarding case fatality risk, older age, higher creatinine levels, and increased pulse rate correlated with higher case fatality risks. Higher GCS scores, systolic blood pressure (SDP), DBP, and oxygen saturation (SpO2) were associated with reduced case fatality risks (Figure 4B,C). We have included a Appendix A to explicitly show the actual data and statistics of all entered variables from our study.

### 3.5. Predictive Value of PNI for 90-Day Case Fatality in Patients with Spontaneous ICH

In the results of a Kaplan–Meier survival curve analysis, the study population was divided into a high PNI group (≥42.77) and a low PNI group (<42.77), showing that the case fatality at 90 days for all causes in the low PNI group was significantly higher than in the high group (log-rank *p* < 0.001; Figure 5).

The Prognostic Nutritional Index (PNI) was evaluated using the ROC curve to predict in-hospital complications and 28-day and 90-day case fatality in patients with spontaneous intracerebral hemorrhage (ICH). The ROC analysis demonstrated that PNI had a moderate ability to predict in-hospital complications with an AUC of 0.575 (95% CI: 0.547–0.604) (Figure 6A). Additionally, time-dependent ROC curves for the prediction of 28-day and 90-day case fatality showed AUC values of 0.637 (95% CI: 0.597–0.676) and 0.645 (95% CI: 0.607–0.683), respectively *(*Figure 6B,C). The sensitivity and specificity at the optimal cut-off point of 42.77 for in-hospital complications were 58% and 53%, respectively. For 28-day and 90-day mortality, the sensitivities were 68% for both periods, while the specificities were 52% each. These findings underscore the value of PNI as a prognostic factor for assessing the risk of in-hospital complications and case fatality in patients with spontaneous ICH.

## 4. Discussion

To the best of our knowledge, this is the first study to compare all-cause in-hospital complications and 28- and 90-day case fatality between high and low PNI groups with patients with spontaneous ICH. The analyses highlighted that the patients with a lower PNI (<42.77) had worse clinical profiles and higher risks of adverse outcomes, including in-hospital complications and higher case fatality rates within 28 and 90 days. Univariate and multivariate logistic regression analyses further substantiated that a higher PNI was associated with a significantly reduced risk of in-hospital complications. Additionally, Cox regression analysis indicated that a higher PNI substantially decreased the risk of case fatality at 28 and 90 days, emphasizing PNI’s role not only as a predictor of immediate complications but also of longer-term case fatality.

After intracerebral hemorrhage, neuroinflammation significantly damages brain tissue [16,17,18] and may lead to immunodepression, which is linked to poor clinical outcomes [17,18,19,20]. Recent evidence indicates that post-ICH neuroinflammation is a major contributor to secondary-induced brain damage and is often associated with a poor clinical prognosis [19,20]. In recent decades, blood-based biomarkers measured upon admission have shown great potential for the diagnosis and prognosis of ICH [10,16,21,22,23]. Stroke-induced immunosuppression decreases immune capacity as lymphocyte numbers reduce, which increases the risk of infection and is strongly associated with an unfavorable functional outcome [24,25]. Another study reported that patients with ICH and lower lymphocyte counts are associated with poor outcomes [23].

Furthermore, malnutrition is associated with impaired immunological function and can increase the risk of infections and case fatality in critically ill patient [26,27]. Previous reports suggested a relationship between hypoalbuminemia and increased case fatality in patients with ICH [9,12]. Typically regarded as a marker of poor nutritional status, hypoalbuminemia also reflects a broader inflammatory response within the body. This condition is a part of a complex acute systemic inflammatory response, suggesting that albumin levels could be depressed not solely by malnutrition but also by the cytokine-mediated response to inflammation. These interactions affect various physiological systems, ultimately impacting the outcomes and survival rates of patients with ICH [28]. Rui et al. showed that a poor-outcome ICH group had worse nutritional and inflammatory statuses [23]. Thus, the prognosis of patients with ICH is related not only to inflammation levels but also to nutritional status. Consequently, PNI combined with serum albumin and lymphocytes can well predict the efficacy of spontaneous ICH, and it can be easily calculated from parameters that are routinely measured in blood parameters at admission.

In this study, we confirmed that the Prognostic Nutritional Index (PNI) is an independent prognostic factor for predicting complications and overall survival in patients with spontaneous intracerebral hemorrhage (ICH). Our findings indicate that a low PNI score correlates with a higher risk of in-hospital complications, as well as increased 28-day and 90-day case fatality rates. Initially, a univariate logistic analysis was conducted to evaluate the association of in-hospital complications with PNI, categorized by a median value of 42.77 (OR = 1). Results demonstrated that a PNI of ≥42.77 significantly reduced the risk of in-hospital complications (OR = 0.64; 95% CI 0.53–0.77; *p* < 0.001). Further exploration through multivariate logistic regression confirmed that a PNI of ≥42.77 independently decreased the risk for in-hospital complications (OR = 0.69, 95% CI: 0.55–0.87, *p* = 0.002). To address potential residual confounding factors, propensity score matching (PSM) analysis was performed, confirming a significant association between lower PNI scores and increased in-hospital complications after matching. Importantly, our results underscore PNI as an independent predictor of complications and 30- and 90-day case fatality in patients with spontaneous ICH. This highlights the clinical utility of PNI in managing and assessing these patients.

In the present study, a low GCS level, anemia, low blood pressure, high fever, and NG insertion have been shown to have a higher risk of in-hospital complications. Langmore et al. identified the need for tube feeding as a predictor of pneumonia, because patients cannot tolerate oral feeding due to impaired consciousness or severe dysphagia [29,30,31]. One meta-analysis showed that anemia at admission was associated with increased case fatality and an increased risk of poor outcomes in patients with ICH [32]. Schwarz S reports that the incidence of fever in patients with hemorrhagic stroke is high, and the duration of fever is associated with poor outcome [33]. The findings of our study are consistent with the previous literature.

Based on the findings of the study, it is important to take action to minimize the nutritional and immune decline in patients with ICH. The early identification of potentially modified risk factors can provide an alternative approach to reduce the risk of complications and improve the outcome. PNI scores can better reflect the balance between nutrition and inflammation than single markers.

We identified some limitations in our study: First, this is a retrospective cohort study conducted in a single center, which leads to inevitable selection bias. In this regard, we adjusted various variables to enhance the accuracy of the results. More validation through prospective multicenter studies is necessary to corroborate our findings. Second, the PNI was evaluated only at admission, neglecting the changes in the laboratory over time. The potential of dynamic PNI to predict the poor prognosis of patients with spontaneous ICH requires further prospective studies. Third, our findings, based on data from Taiwan, may not be generalizable to other populations. International studies are necessary to validate our results. Fourth, the exclusion of nearly 50% of initially enrolled patients introduces potential bias in the final cohort, which may affect the study’s conclusion.

## 5. Conclusions

This retrospective study revealed that PNI may be a valuable marker to predict the outcomes of patients with spontaneous ICH. Our findings suggest that a lower PNI is associated with increased in-hospital complications and higher case fatality rates at 28 and 90 days. Additionally, the study suggests that PNI, reflecting the intersection of nutritional and inflammatory statuses, could serve as a more comprehensive prognostic factor compared to traditional single markers.

## Figures and Tables

**Figure 1 nutrients-16-01841-f001:**
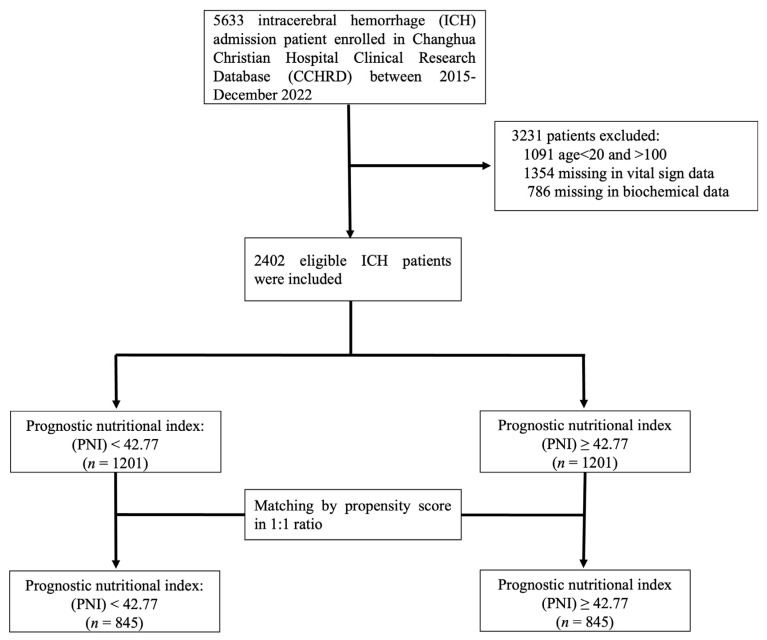
Flowchart of patient selection.

**Figure 2 nutrients-16-01841-f002:**
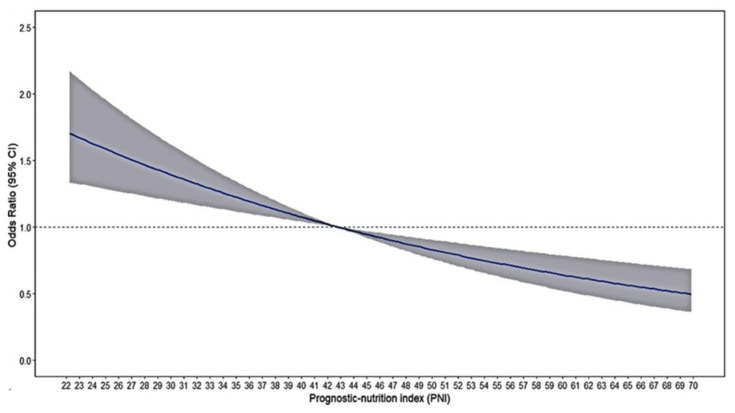
Restricted cubic spline regression between associations of PNI levels with in-hospital complication. The odds ratio (OR) is represented by the solid line, and the 95% confidence interval (CI) by the shaded area. Dotted line was that there is no association between the PNI and the odds of in-hospital complication.

**Figure 3 nutrients-16-01841-f003:**
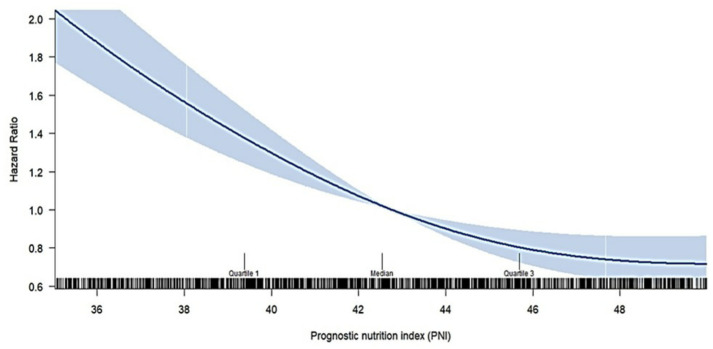
Association between PNI and the risk of 90-day all-cause case fatality using a restricted cubic spline regression model in the total population. The hazard ratios (HRs) are represented by the solid line and the 95% confidence interval (CI) by the shaded area. White line in plot is represented ad quartile of data.

**Figure 4 nutrients-16-01841-f004:**
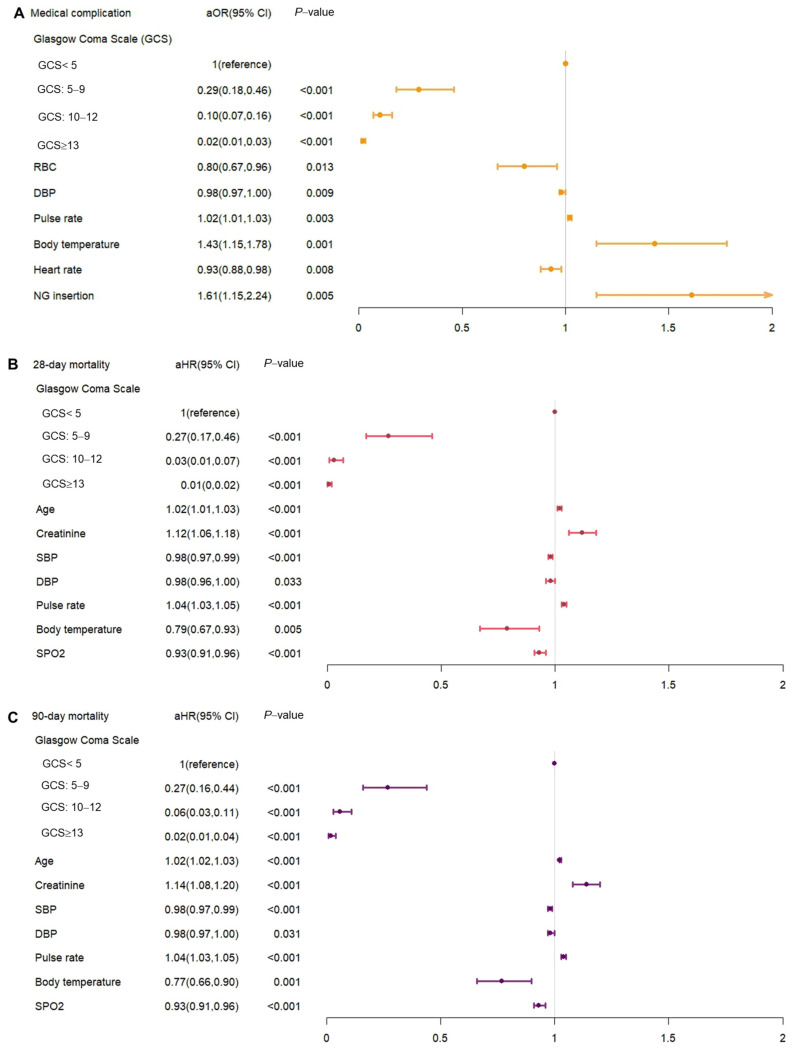
The significant risk factor for (**A**) medical complication, (**B**) 28-day case fatality, and (**C**) 90-day case fatality. GCS, Glasgow Coma Scale; DBP, diastolic blood pressure; SBP, systolic blood pressure; SpO2, percutaneous oxygen saturation; RBC, red blood cell. All multivariate regression models were adjusted for variables including sex, age, GCS scores, history of comorbidities (DM, hypertension, hyperlipidemia, and Charlson comorbidity score), vital signs, and lab data (RBC, WBC, Hb, platelet count, creatinine, BUN, K, Na, and HbA1c). Dots in plot are represented odds ratio.

**Figure 5 nutrients-16-01841-f005:**
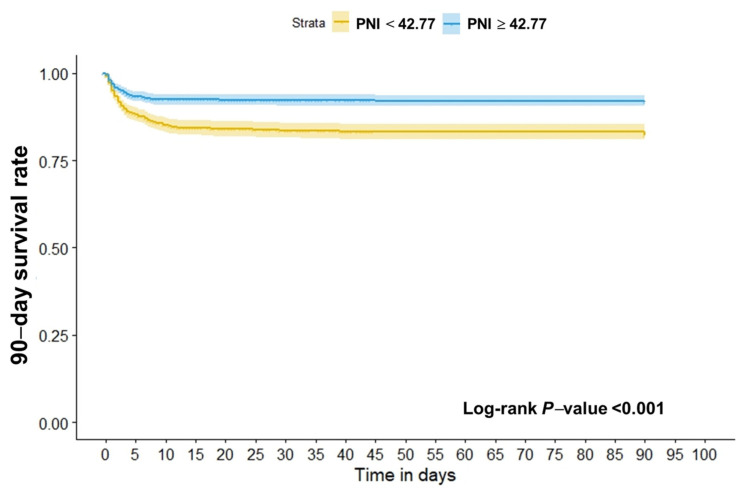
Kaplan–Meier curve for 90-day case fatality in relation to PNI estimated using restricted cubic spline models. The survival curve is represented by the solid line and the 95% confidence interval (CI) by the shaded area.

**Figure 6 nutrients-16-01841-f006:**
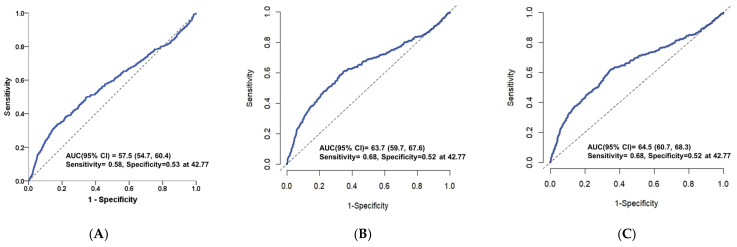
(**A**) Receiver operating characteristics curve (ROC) of the PNI for prediction of in-hospital complications. (**B**) Time-dependent ROC curve for prediction of case fatality at 28 days. (**C**) Time-dependent ROC curve for prediction of case fatality at 90 days. Dashed line was the reference line, which represented ROC at 0.5.

**Table 1 nutrients-16-01841-t001:** The demographics and clinical characteristics divided by the cut-off value of PNI score = 42.77 before and after the propensity score matching.

Variables	Before Propensity Score Matching *n* = 2402				After Propensity Score Matching *n* = 1690		
	Total	PNI < 42.77	PNI ≥ 42.77	*p*-Value	PNI < 42.77	PNI ≥ 42.77	*p*-Value
Characteristics number	2402	1201	1201		845	845	
Age (years)	63.6 ± 18.6	67.3 ± 17.6	59.9 ± 18.7	<0.001	63.6 ± 17.9	64.4 ± 17.3	0.345
Gender, male, *n* (%)	1419 (59.1%)	705 (58.7%)	714 (59.5%)	0.709	492 (58.2%)	496 (58.7%)	0.843
BMI	23.9 ± 7.5	23.5 ± 9.8	24.4 ± 4.2	0.004	23.6 ± 4.2	24 ± 4.1	0.031
GCS at admission [median (IQR)]	13 (9, 14)	13 (7, 14)	14 (10, 14)	<0.001	13 (8, 14)	13 (10, 14)	0.044
Vital sign at admission							
SBP (mmHg)	131 ± 16.7	132.4 ± 16.7	129.6 ± 16.6	<0.001	131.5 ± 16.3	131.7 ± 16.5	0.865
DBP (mmHg)	72.9 ± 10.8	72.5 ± 11	73.4 ± 10.7	0.039	73.5 ± 11	73.4 ± 10.8	0.841
Heart rate (bpm)	80.1 ± 14.3	81.9 ± 14.9	78.3 ± 13.6	<0.001	79.8 ± 14.3	79.8 ± 13.9	0.943
Body temperature (°C)	36.6 ± 0.5	36.5 ± 0.6	36.6 ± 0.5	0.023	36.6 ± 0.6	36.6 ± 0.5	0.806
Respiratory rate (bpm)	17.8 ± 2.3	17.9 ± 2.5	17.7 ± 2.1	0.006	17.8 ± 2.2	17.8 ± 2.2	0.694
SpO2 (%)	97.7 ± 2.1	97.7 ± 2	97.7 ± 2.1	0.591	97.7 ± 2.1	97.7 ± 1.9	0.933
Laboratory data at admission							
Creatinine (mg/dl)	1.1 ± 1.3	1.3 ± 1.5	1 ± 1	<0.001	1.1 ± 1.3	1.1 ± 1.1	0.804
Hb (g/dl)	13 ± 2.1	12.3 ± 2.1	13.6 ± 1.9	<0.001	12.8 ± 2	13.3 ± 1.9	<0.001
K (mmol/l)	3.8 ± 0.5	3.8 ± 0.6	3.7 ± 0.5	0.020	3.8 ± 0.5	3.8 ± 0.5	0.876
Na (mmol/l)	137.2 ± 3.8	136.7 ± 4.2	137.7 ± 3.1	<0.001	137.4 ± 3.5	137.3 ± 3.3	0.587
BUN (mg/dl)	18.3 ± 12.1	19.5 ± 13.5	17 ± 10.4	<0.001	17.8 ± 11.4	17.9 ± 11.9	0.779
RBC (10 × 6/μL )	4.3 ± 0.8	4.1 ± 0.8	4.5 ± 0.7	<0.001	4.3 ± 0.7	4.3 ± 0.7	0.291
RDW (%)	14.1 ± 1.7	14.4 ± 1.9	13.8 ± 1.3	<0.001	14.2 ± 1.7	13.9 ± 1.4	<0.001
HbA1c (%)	6.3 ± 1.1	6.2 ± 1.1	6.3 ± 1	0.246	6.3 ± 1.1	6.3 ± 1	0.647
Albumin (g/dl)	3.4 ± 0.5	3.2 ± 0.4	3.7 ± 0.4	<0.001	3.2 ± 0.4	3.7 ± 0.4	<0.001
Platelet (10 × 3/μL)	208.5 ± 79.8	199.7 ± 85.9	217.2 ± 72.2	<0.001	206.5 ± 85	211.3 ± 74	0.217
WBC (10 × 3/μL)	10.1 ± 4.8	10.2 ± 5.1	10 ± 4.5	0.420	10.5 ± 5.3	9.8 ± 4.3	0.001
Lymphocyte (%)	19.6 ± 11.9	13.3 ± 8.6	25.9 ± 11.4	<0.001	13.3 ± 8.2	25.9 ± 11.4	<0.001
Charlson Comorbidity Index	1.9 ± 2.6	2.3 ± 2.9	1.6 ± 2.3	<0.001	2 ± 3	2 ± 2	0.514
Medical history							
Antihypertension treatment, *n* (%)	795 (33.1%)	443 (36.9%)	352 (29.3%)	<0.001	282 (33.4%)	293 (34.7%)	0.572
Lipid-lowering treatment, *n* (%)	331 (13.8%)	175 (14.6%)	156 (13%)	0.261	120 (14.2%)	123 (14.6%)	0.835
Non-insulin antihyperglycemic treatment, *n* (%)	281 (11.7%)	157 (13.1%)	124 (10.3%)	0.036	90 (10.7%)	96 (11.4%)	0.641
Insulin treatment, *n* (%)	125 (5.2%)	80 (6.7%)	45 (3.7%)	0.001	36 (4.3%)	40 (4.7%)	0.639
NG insertion during admission, *n* (%)	409 (17%)	211 (17.6%)	198 (16.5%)	0.480	140 (16.6%)	133 (15.7%)	0.644
Propensity score	0.5 ± 0.2	0.4 ± 0.2	0.6 ± 0.2	<0.001	0.5 ± 0.1	0.5 ± 0.1	0.962

Data are presented as mean ± SD, *n*, median [IQR], or *n* (%). BMI, body mass index; GCS, Glasgow Coma Scale; SBP, systolic blood pressure; DBP, diastolic blood pressure; SpO2, percutaneous oxygen saturation; RBC, red blood cell; Hb, hemoglobin; WBC, white blood cell; RDW, red blood cell distribution width; BUN, blood urea nitrogen; HbA1c, Glycohemoglobin; NG tube, nasogastric tube; SD, standard deviation; and IQR, interquartile range.

**Table 2 nutrients-16-01841-t002:** Univariate and multivariate analyses of in-hospital complication in the PNI group using before and after propensity score-matched datasets.

	Univariate Analysis		Multivariate Analysis		Propensity Score Match Analysis	
	Crude OR (95% CI)	*p*-Value	Adjusted OR (95% CI)	*p*-Value	PSM OR (95% CI)	*p*-Value
Overall complication						
PNI < 42.77	1 (reference)		1 (reference)		1 (reference)	
PNI ≥ 42.77	0.64 (0.53, 0.77)	<0.001	0.69 (0.55, 0.87)	0.002	0.74 (0.58, 0.94)	0.013

PSM, propensity score matching; OR, odds ratio; CI, confidence interval. All multivariate regression models were adjusted for variables: sex, age, GCS scores, history of diabetes mellitus, hypertension, and hyperlipidemia, temperature, heart rate, respiratory rate, systolic blood pressure (SBP), diastolic blood pressure (DBP), percutaneous oxygen saturation (SpO2), red blood cells (RBCs), white blood cells (WBCs), lymphocytes, hemoglobin concentration, platelet count, serum creatinine, blood urea nitrogen (BUN), serum potassium, serum sodium, HbA1c, and Charlson Comorbidity Index.

**Table 3 nutrients-16-01841-t003:** Univariate and multivariate Cox proportional hazards regression analysis of PNI level affecting overall survival before and after propensity score matching.

	Univariate Analysis		Multivariate Analysis		Propensity Score Match Analysis	
	Crude HR (95% CI)	*p*-Value	Adjusted HR (95% CI)	*p*-Value	PSM HR (95% CI)	*p*-Value
28-day case fatality						
PNI < 42.77	1 (reference)		1 (reference)		1 (reference)	
PNI ≥ 42.77	0.46 (0.36, 0.59)	<0.001	0.72 (0.55, 0.94)	0.015	0.71(0.52, 0.98)	0.039
90-day case fatality						
PNI < 42.77	1 (reference)		1 (reference)		1(reference)	
PNI ≥ 42.77	0.45 (0.35, 0.57)	<0.001	0.72 (0.56, 0.93)	0.011	0.69 (0.51, 0.94)	0.020

PSM, propensity score matching: a multivariate regression model; HR, hazard ratio; CI, confidence interval. All multivariate regression models were adjusted for variables: sex, age, GCS scores, history of diabetes mellitus, hypertension, and hyperlipidemia, temperature, heart rate, respiratory rate, systolic blood pressure (SBP), diastolic blood pressure (DBP), percutaneous oxygen saturation (SPO2), red blood cells (RBCs), white blood cells (WBCs), lymphocytes, hemoglobin concentration, platelet count, serum creatinine, blood urea nitrogen (BUN), serum potassium, serum sodium, HbA1c, and Charlson Comorbidity Index.

## Data Availability

Data are contained within the article.

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
