# Peer review of "Low Prognostic Nutritional Index Predicts In-Hospital Complications and Case Fatality in Patients with Spontaneous Intracerebral Hemorrhage: A Retrospective Study"

_nutrients, 2024, doi:10.3390/nu16121841_

Round 1

Reviewer 1 Report

Comments and Suggestions for Authors

This is an interesting study on the significance of PNI as a potential marker for prognosis in ICH. While this are some interesting insights, i have several suggestions for author to consider;

a. Abstract: In the background, please add sentences on the importance of nutritional status in outcomes of neurological conditions like ICH. Also, please state in the Methods: patient population that were included vis a vis (inclusion/exclusion criteria)

3. Introduction: Suggest clarfying the main objectives or hypotheses of the study.

4. Methods, It is not clear why the baseline stroke severity and pre-morbid functional status were not considered? Besides, age>80, and ICH volume of >=30 ml and intraventriculr hemorrhage are important predictors of outcomes in these patients. It is not clear why these variables were not included in the modelling.

5. There are several grammatical errors and poor comprehension that makes for difficulty in reading. suggest the authors improve the clarity and flow of the manuscript.

6. In light of the variables discussed above, it is not clear how the current model may be appropriate for prognostic modelling. 

7. The limitations section also needs further expansion. Suggest adding that the there are limitations associated with using PNI as a predictor, given its reliance on a single assessment at admission and poor ability to capture dynamic changes in patients' nutritional and immune status over time.

8. Suggest breaking down long sentences to smaller parts. 

9. Suggest providing discussion for opportunities for future research, such as investigating the feasibility of incorporating dynamic PNI assessments into clinical practice as well as the need for independent validation across diverse populations.

10. In the Discussion, potential mechanisms underlying associations between PNI scores and outcomes in spontaneous ICH patients can also be discussed. This can help readers gain a better understanding on the influence of nutrition and inflammation on prognosis.

Comments on the Quality of English Language

Suggest breaking down long sentences to smaller parts. There are several grammatical errors and poor comprehension that makes for difficulty in reading. suggest the authors improve the clarity and flow of the manuscript.

Reviewer 2 Report

Comments and Suggestions for Authors

This paper reports on a single centre retrospective cohort study of the association between the prognostic nutrition index (PNI) and in-hospital complication and mortality in patients with spontaneous intracerebral haemorrhage (ICH). The authors found that PNI could serve as a valuable marker for predicting medical complications and mortality in patients with spontaneous ICH.

There are a number of issues the authors may wish to attend to:

1.       Line 21 – the PNI level selected is better placed in Results part of the Abstract

2.       Line 24 – ‘which could reduce potential bias to some extent.’ can be deleted

3.       Line 25 – please provide some patient demographics and overall results first

4.       Line 36 – ‘In Taiwan..’ and ‘..worldwide..’  is not compatible…

5.       Line 38 – why selectively mention US data? Why not from a more global study?

6.       Line 41 – please provide the reference

7.       Line 59  - ‘..immunological nutritional..’ is better stated as ‘..nutritional and immunological..’

8.       Line 108 (major) – what is the rationale for choosing 42.77 as the cut-off point?

9.       Lines 125, 126-127 – references?

10.   Lines 138-144 – any were already mentioned earlier in lines 98-101 – please place all in 1 part of this section

11.   Results (major) – how were the excluded patients different from the included patients?

12.   Line 171 – use ‘,’ in place of ‘.’

13.   Line 187-188 (major) - this crucial information should have come much earlier

14.   Tables 2 and 3, Fig 4 A B C  (major)– please show the data of all entered variables

15.   Line 265 (major) – what is the sensitivity and specificity?

16.   Lines 271-274 – why repeated from the previous paragraph?

17.   Discussion (major) – please start with a qualitative summary of the main findings

18.   Line 287 – ‘After ICH…” can be a new paragraph

19.   Line 325 - ‘who’ can be deleted

20.   Limitation (major) – also 50% of the patients were excluded; please think of more limitations

21.   Conclusions – too short…

Comments on the Quality of English Language

Some attention needed

Reviewer 3 Report

Comments and Suggestions for Authors Jhang et al are presenting an interesting study aiming to compare the association of Nutritional Index with ouctomes of stroke patients. I find the study very interesiting and of scientific  interest. I do have the following points:

      â€‹1. "n Taiwan, cerebrovascular diseases are the leading cause of death worldwide and 36 have been ranked fifth among the top ten causes of death in 2022 [1]. Approximately 10% 37 of the 795,000 strokes that occur annually in the United States are intracerebral hemor- 38 rhages (ICH)[1]. â€‹"   - please clarify if the statistics are of Taiwan or global

2. "paproximately 10%  of the 795,000 strokes that occur annually in the United States are intracerebral hemor- 38 rhages (ICH)[1]" - replace USA data by worldwide  data on incidence or prevalence of ICH

3." The incidence of ICH incidence is higher in Taiwan than 41 in Western population" - rephrase this sentence 

​4. "Previous research has also indicated that 49 acute ICH patient with poor outcomes had a greater degree of undernutrition compared 50 to those without (p<0.001)" - no need to  include specific statical figure

5. "enrolling all patients admitte" -  replace enrolling (to be used in active prospective inclusion) by including  

6. What is the rationale for exclusion of patients> 100 years ?

7. No need to explain what is the meaning and how is calculated  the Glasgow coma scale

8. Table 2 and 3 should contain the statiscal figures (OR) of the variables including the multivariable regression analysis

9. In the entire manuscript: replace mortality by case-fatality

10. "In this study, we confirmed that the PNI score was an independent biomarker to predict complications and overall survival"  PNI cant be considered a biomarker (it is not a biological molecule or a specific individual physiological or imaging finding)

11. The discussion should be enriched with data from other studies. Hypoalbuminemia for instance is associate with  systemic inflammatory response (DOI: 10.1016/j.jcrc.2017.06.002) 

Round 2

Reviewer 1 Report

Comments and Suggestions for Authors No further comments. I recommend the manuscript be accepted.

Comments on the Quality of English Language

/

Reviewer 2 Report

Comments and Suggestions for Authors

This is a revised submission of a paper reports on a single centre retrospective cohort study of the association between the prognostic nutrition index (PNI) and in-hospital complication and mortality in patients with spontaneous intracerebral haemorrhage (ICH). The paper is much improved.

There are a small number of issues mentioned earlier that the authors have not yet attended to:

1.       Line 26 – again, please provide some patient demographics and overall results first eg age and sex of participants, overall complications mortality and PNI

2.       Results (major) – again, how were the excluded patients different from the included patients? Best to provide a table comparing included and excluded patients based on available data eg demographics, risk factors, etc, and show if there are any statistical differences  

3.       Tables 2 and 3, Fig 4 A B C (major) – again, please show the actual data of ALL entered variables and the statistics
